# THE IMPLICIT BIAS OF DEPTH: HOW INCREMENTAL LEARNING DRIVES GENERALIZATION

**Daniel Gissin, Shai Shalev-Shwartz, Amit Daniely**
School of Computer Science
The Hebrew University
Jerusalem, Israel
`{daniel.gissin,shais,amit.daniely}@mail.huji.ac.il`

## ABSTRACT

A leading hypothesis for the surprising generalization of neural networks is that the dynamics of gradient descent bias the model towards simple solutions, by searching through the solution space in an incremental order of complexity. We formally define the notion of incremental learning dynamics and derive the conditions on depth and initialization for which this phenomenon arises in deep linear models. Our main theoretical contribution is a dynamical depth separation result, proving that while shallow models can exhibit incremental learning dynamics, they require the initialization to be exponentially small for these dynamics to present themselves. However, once the model becomes deeper, the dependence becomes polynomial and incremental learning can arise in more natural settings. We complement our theoretical findings by experimenting with deep matrix sensing, quadratic neural networks and with binary classification using diagonal and convolutional linear networks, showing all of these models exhibit incremental learning.

## 1 INTRODUCTION

Neural networks have led to a breakthrough in modern machine learning, allowing us to efficiently learn highly expressive models that still generalize to unseen data. The theoretical reasons for this success are still unclear, as the generalization capabilities of neural networks defy the classic statistical learning theory bounds. Since these bounds, which depend solely on the capacity of the learned model, are unable to account for the success of neural networks, we must examine additional properties of the learning process. One such property is the optimization algorithm - while neural networks can express a multitude of possible ERM solutions for a given training set, gradient-based methods with the right initialization may be implicitly biased towards certain solutions which generalize.

A possible way such an implicit bias may present itself, is if gradient-based methods were to search the hypothesis space for possible solutions of gradually increasing complexity. This would suggest that while the hypothesis space itself is extremely complex, our search strategy favors the simplest solutions and thus generalizes. One of the leading results along these lines has been by Saxe et al. (2013), deriving an analytical solution for the gradient flow dynamics of deep linear networks and showing that for such models, the singular values converge at different rates, with larger values converging first. At the limit of infinitesimal initialization of the deep linear network, Gidel et al. (2019) show these dynamics exhibit a behavior of "incremental learning" - the singular values of the model are learned separately, one at a time. Our work generalizes these results to small but finite initialization scales.

Incremental learning dynamics have also been explored in gradient descent applied to matrix completion and sensing with a factorized parameterization (Gunasekar et al. (2017), Arora et al. (2018), Woodworth et al. (2019)). When initialized with small Gaussian weights and trained with a small learning rate, such a model is able to successfully recover the low-rank matrix which labeled the data, even if the problem is highly over-determined and no additional regularization is applied. In their proof of low-rank recovery for such models, Li et al. (2017) show that the model remains low-rank throughout the optimization process, leading to the successful generalization. Additionally,

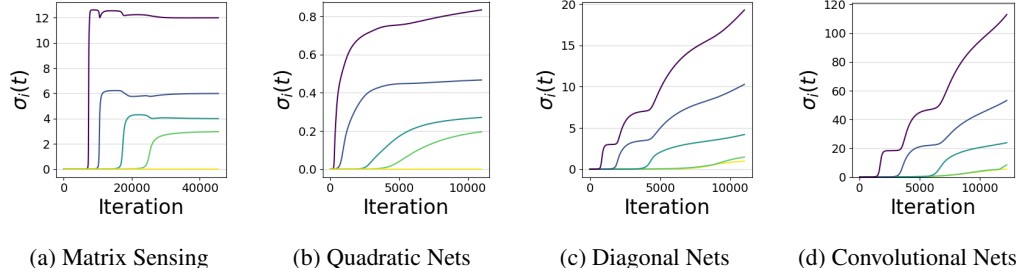

|     (a) Matrix Sensing | (b) Quadratic Nets | (c) Diagonal Nets | (d) Convolutional Nets |

Figure 1: Incremental learning dynamics in deep models. Each panel shows the evolution of the five largest values of $\sigma$, the parameters of the induced model. All models were trained using gradient descent with a small initialization and learning rate, on a small training set such that there are multiple possible solutions. In all cases, the deep parameterization of the models lead to "incremental learning", where the values are learned at different rates (larger values are learned first), leading to sparse solutions. (a) Depth 4 matrix sensing, $\sigma$ denotes singular values (see section 4.1). (b) Quadratic networks, $\sigma$ denotes singular values (see section 4.2). (c) Depth 3 diagonal networks, $\sigma$ denotes feature weights (see section 4.3). (d) Depth 3 circular-convolutional networks, $\sigma$ denotes amplitudes in the frequency domain of the feature weights (see appendix G).

Arora et al. (2019) explore the dynamics of such models, showing the singular values are learned at different rates and that deeper models exhibit stronger incremental learning dynamics. Our work deals with a more simplified setting, allowing us to determine explicitly under which conditions depth leads to this dynamical phenomenon.

Finally, the learning dynamics of nonlinear models have been studied as well. Combes et al. (2018) and Williams et al. (2019) study the gradient flow dynamics of shallow ReLU networks under restrictive distributional assumptions, Ronen et al. (2019) show that shallow networks learn functions of gradually increasing frequencies and Nakkiran et al. (2019) show how deep ReLU networks correlate with linear classifiers in the early stages of training.

These findings, along with others, suggest that the generalization ability of deep networks is at least in part due to the incremental learning dynamics of gradient descent. Following this line of work, we begin by explicitly defining the notion of incremental learning for a toy model which exhibits this sort of behavior. Analyzing the dynamics of the model for gradient flow and gradient descent, we characterize the effect of the model's depth and initialization scale on incremental learning, showing how deeper models allow for incremental learning in larger (realistic) initialization scales. Specifically, we show that a depth-2 model requires exponentially small initialization for incremental learning to occur, while deeper models only require the initialization to be polynomially small.

Once incremental learning has been defined and characterized for the toy model, we generalize our results theoretically and empirically for larger linear and quadratic models. Examples of incremental learning in these models can be seen in figure 1, which we discuss further in section 4.

## 2 DYNAMICAL ANALYSIS OF A TOY MODEL

We begin by analyzing incremental learning for a simple model. This will allow us to gain a clear understanding of the phenomenon and the conditions for it, which we will later be able to apply to a variety of other models in which incremental learning is present.

### 2.1 PRELIMINARIES

Our simple linear model will be similar to the toy model analyzed by Woodworth et al. (2019). Our input space will be $\mathcal{X} = \mathbb{R}^d$ and the hypothesis space will be linear models with non-negative weights, such that:

$$f_\sigma(x) = \langle \sigma, x \rangle \qquad \sigma \in \mathbb{R}^d_{\geq 0} \qquad (1)$$

We will introduce depth into our model, by parameterizing $\sigma$ using $w \in \mathbb{R}^d_{\geq 0}$ in the following way:

$$\forall i : \ \sigma_i = w_i^N$$

Where $N$ represents the depth of the model. Since we restrict the model to having non-negative weights, this parameterization doesn't change the expressiveness, but it does radically change it's optimization dynamics.

Assuming the data is labeled by some $\sigma^* \in \mathbb{R}^d_{\geq 0}$, we will study the dynamics of this model for general $N$ under a depth-normalized[1] squared loss over Gaussian inputs, which will allow us to derive our analytical solution:

$$\ell_N(w) = \frac{1}{2N^2} \mathbb{E}_x[(\langle \sigma^*, x \rangle - \langle w^N, x \rangle)^2] = \frac{1}{2N^2} ||w^N - \sigma^*||^2 \tag{2}$$

We will assume that our model is initialized uniformly with a tunable scaling factor, such that:

$$\forall i : \ w_i(0) = \sqrt[N]{\sigma_0} \tag{3}$$

## 2.2 GRADIENT FLOW ANALYTICAL SOLUTIONS

Analyzing our toy model using gradient flow allows us to obtain an analytical solution for the dynamics of $\sigma(t)$ along with the dynamics of the loss function for a general $N$. For brevity, the following theorem refers only to $N = 1, 2$ and $N \to \infty$, however the solutions for $3 \leq N < \infty$ are similar in structure to $N \to \infty$, but more complicated. We also assume $\sigma_i^* > 0$ for brevity, however we can derive the solutions for $\sigma_i^* = 0$ as well. Note that this result is a special case adaptation of the one presented in Saxe et al. (2013) for deep linear networks:

**Theorem 1.** *Minimizing the toy linear model described in* (1) *with gradient flow over the depth normalized squared loss* (2), *with Gaussian inputs and weights initialized as in* (3) *and assuming $\sigma_i^* > 0$ leads to the following analytical solutions for different values of $N$:*

$$N = 1: \quad \sigma_i(t) = \sigma_i^* + (\sigma_0 - \sigma_i^*)e^{-t}$$

$$N = 2: \quad \sigma_i(t) = \frac{\sigma_0 \sigma_i^* e^{\sigma_i^* t}}{\sigma_0(e^{\sigma_i^* t} - 1) + \sigma_i^*}$$

$$N \to \infty: \quad t = \frac{1}{(\sigma_i^*)^2} \log \left( \frac{\sigma_i(t)(\sigma_0 - \sigma_i^*)}{\sigma_0(\sigma_i(t) - \sigma_i^*)} \right) - \frac{1}{\sigma_i^*} \left( \frac{1}{\sigma_i(t)} - \frac{1}{\sigma_0} \right)$$

*Proof.* The gradient flow equations for our model are the following:

$$\dot{w}_i = -\nabla_{w_i} \ell = \frac{1}{N} w_i^{N-1}(\sigma_i^* - w_i^N)$$

Given the dynamics of the $w$ parameters, we may use the chain rule to derive the dynamics of the induced model, $\sigma$:

$$\dot{\sigma}_i = \frac{d\sigma_i}{dw_i} \dot{w}_i = w^{2N-2}(\sigma_i^* - w_i^N) = \sigma_i^{2 - \frac{2}{N}}(\sigma_i^* - \sigma_i) \tag{4}$$

This differential equation is solvable for all $N$, leading to the solutions in the theorem. Taking $N \to \infty$ in (4) leads to $\dot{\sigma}_i = \sigma_i^2(\sigma_i^* - \sigma_i)$, which is also solvable.

$\square$

---

[1]This normalization is used for mathematical convenience to have solutions of different depths exhibit similar time scales in their dynamics. Equivalently, we can derive the solutions for the regular square loss and then use different time scalings in the dynamical analysis.

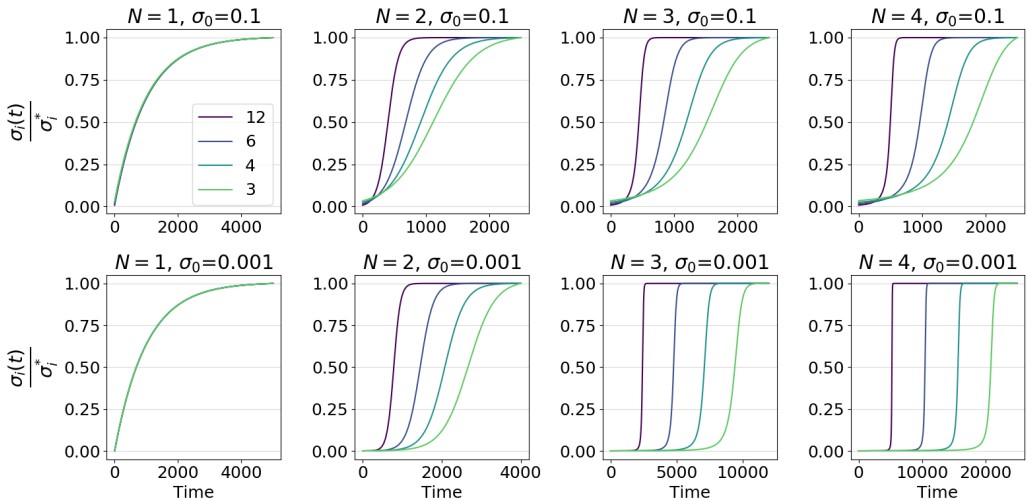

Figure 2: Incremental learning dynamics in the toy model. Each panel shows the evolution of $\frac{\sigma_i(t)}{\sigma_i^*}$ for $\sigma_i^* \in \{12, 6, 4, 3\}$ according to the analytical solutions in theorem 1, under different depths and initializations. The first column has all values converging at the same rate. Notice how the deep parameterization with small initialization leads to distinct phases of learning, where values are learned incrementally (bottom-right). The shallow model's much weaker incremental learning, even at small initialization scales (second column), is explained in theorem 2.

Analyzing these solutions, we see how even in such a simple model depth causes different factors of the model to be learned at different rates. Specifically, values corresponding to larger optimal values converge faster, suggesting a form of incremental learning. This is most clear for $N = 2$ where the solution isn't implicit, but is also the case for $N \geq 3$, as we will see in the next subsection.

These dynamics are depicted in figure 2, where we see the dynamics of the different values of $\sigma(t)$ as learning progresses. When $N = 1$, all values are learned at the same rate regardless of the initialization, while the deeper models are clearly biased towards learning the larger singular values first, especially at small initialization scales.

Our model has only one optimal solution due to the population loss, but it is clear how this sort of dynamic can induce sparse solutions - if the model is able to fit the data after a small amount of learning phases, then it's obtained result will be sparse. Alternatively, if $N = 1$, we know that the dynamics will lead to the minimal $\ell_2$ norm solution which is dense. We explore the sparsity inducing bias of our toy model by comparing it empirically[2] to a greedy sparse approximation algorithm in appendix D, and give our theoretical results in the next section.

## 3    INCREMENTAL LEARNING

Equipped with analytical solutions for the dynamics of our model for every depth, we turn to study how the depth and initialization effect incremental learning. While Gidel et al. (2019) focuses on incremental learning in depth-2 models at the limit of $\sigma_0 \to 0$, we will study the phenomenon for a general depth and for $\sigma_0 > 0$.

First, we will define the notion of incremental learning. Since all values of $\sigma$ are learned in parallel, we can't expect one value to converge before the other moves at all (which happens for infinitesimal initialization as shown by Gidel et al. (2019)). We will need a more relaxed definition for incremental learning in finite initialization scales.

---

[2]The code for reproducing all of our experiments can be found in `https://github.com/dsgissin/Incremental-Learning`

**Definition 1.** *Given two values $\sigma_i, \sigma_j$ such that $\sigma_i^* > \sigma_j^* > 0$ and both are initialized as $\sigma_i(0) = \sigma_j(0) = \sigma_0 < \sigma_j^*$, and given two scalars $s \in (0, \frac{1}{4})$ and $f \in (\frac{3}{4}, 1)$, we call the learning of the values $(s, f)$-incremental if there exists a $t$ for which:*

$$\sigma_j(t) \leq s\sigma_j^* < f\sigma_i^* \leq \sigma_i(t)$$

In words, two values have distinct learning phases if the first almost converges ($f \approx 1$) before the second changes by much ($s \ll 1$). Note that for any $N$, $\sigma(t)$ is monotonically increasing and so once $\sigma_j(t) = s\sigma_j^*$, it will not decrease to allow further incremental learning. Given this definition of incremental learning, we turn to study the conditions that facilitate incremental learning in our toy model.

Our main result is a dynamical depth separation result, showing that incremental learning is dependent on $\frac{\sigma_i^*}{\sigma_j^*}$ in different ways for different values of $N$. The largest difference in dependence happens between $N = 2$ and $N = 3$, where the dependence changes from exponential to polynomial:

**Theorem 2.** *Given two values $\sigma_i, \sigma_j$ of a toy linear model as in (1), where $\frac{\sigma_i^*}{\sigma_j^*} = r > 1$ and the model is initialized as in (3), and given two scalars $s \in (0, \frac{1}{4})$ and $f \in (\frac{3}{4}, 1)$, then the largest initialization value for which the learning phases of the values are $(s, f)$-incremental, denoted $\sigma_0^{th}$, is bounded in the following way:*

$$s\sigma_j^* \left( \frac{s}{rf} \right)^{\frac{1}{(1-f)(r-1)}} \leq \sigma_0^{th} \leq s\sigma_j^* \left( \frac{s}{rf} \right)^{\frac{1-s}{r-1}} \qquad N = 2$$

$$s\sigma_j^* \left( \frac{(1-f)(r-1)}{1 + (1-f)(r-1)} \right)^{\frac{N}{N-2}} \leq \sigma_0^{th} \leq s\sigma_j^* \left( \frac{r-1}{r-s} \right)^{\frac{N}{N-2}} \qquad N \geq 3$$

*Proof sketch (the full proof is given in appendix A).* Rewriting the separable differential equation in (4) to calculate the time until $\sigma(t) = \alpha\sigma^*$, we get the following:

$$t_\alpha(\sigma) = \int_{\sigma_0}^{\alpha\sigma^*} \frac{d\sigma}{\sigma^{2-\frac{2}{N}}(\sigma^* - \sigma)}$$

The condition for incremental learning is then the requirement that $t_f(\sigma_i) \leq t_s(\sigma_j)$, resulting in:

$$\int_{\sigma_0}^{f\sigma_i^*} \frac{d\sigma}{\sigma^{2-\frac{2}{N}}(\sigma_i^* - \sigma)} \leq \int_{\sigma_0}^{s\sigma_j^*} \frac{d\sigma}{\sigma^{2-\frac{2}{N}}(\sigma_j^* - \sigma)}$$

We then relax/restrict the above condition to get a necessary/sufficient condition on $\sigma_0$, leading to a lower and upper bound on $\sigma_0^{th}$.

$\square$

Note that the value determining the condition for incremental learning is $\frac{\sigma_i^*}{\sigma_j^*}$ - if two values are in the same order of magnitude, then their ratio will be close to $1$ and we will need a small initialization to obtain incremental learning. The dependence on the ratio changes with depth, and is exponential for $N = 2$. This means that incremental learning, while possible for shallow models, is difficult to see in practice. This result explains why changing the initialization scale in figure 2 changes the dynamics of the $N \geq 3$ models, while not changing the dynamics for $N = 2$ noticeably.

The next theorem extends part of our analysis to gradient descent, a more realistic setting than the infinitesimal learning rate of gradient flow:

**Theorem 3.** *Given two values $\sigma_i, \sigma_j$ of a depth-2 toy linear model as in (1), such that $\frac{\sigma_i^*}{\sigma_j^*} = r > 1$ and the model is initialized as in (3), and given two scalars $s \in (0, \frac{1}{4})$ and $f \in (\frac{3}{4}, 1)$, and assuming $\sigma_j^* \geq 2\sigma_0$, and assuming we optimize with gradient descent with a learning rate $\eta \leq \frac{c}{\sigma_1^*}$ for*

$c < 2(\sqrt{2} - 1)$ *and $\sigma_1^*$ the largest value of $\sigma^*$, then the largest initialization value for which the learning phases of the values are $(s, f)$-incremental, denoted $\sigma_0^{th}$, is lower and upper bounded in the following way:*

$$\frac{1}{2}\frac{s}{1-s}\sigma_j^*\Big(\frac{1-f}{2rf}\frac{s}{1-s}\Big)^{\frac{1}{A-1}} \leq \sigma_0^{th} \leq \frac{s}{1-s}\sigma_j^*\Big(\frac{1-f}{f}\frac{s}{1-s}\Big)^{\frac{1}{B-1}}$$

*Where $A$ and $B$ are defined as:*

$$A = \frac{\log\big(1 - c\frac{\sigma_i^*}{\sigma_1^*} + c^2\big(\frac{\sigma_i^*}{\sigma_1^*}\big)^2\big)}{\log\big(1 - c\frac{\sigma_j^*}{\sigma_1^*} - \frac{c^2}{4}\big(\frac{\sigma_j^*}{\sigma_1^*}\big)^2\big)} \qquad B = \frac{\log\big(1 - c\frac{\sigma_i^*}{\sigma_1^*} - \frac{c^2}{4}\big(\frac{\sigma_i^*}{\sigma_1^*}\big)^2\big)}{\log\big(1 - c\frac{\sigma_j^*}{\sigma_1^*} + c^2\big(\frac{\sigma_j^*}{\sigma_1^*}\big)^2\big)}$$

We defer the proof to appendix B.

Note that this result, while less elegant than the bounds of the gradient flow analysis, is similar in nature. Both $A$ and $B$ simplify to $r$ when we take their first order approximation around $c = 0$, giving us similar bounds and showing that the condition on $\sigma_0$ for $N = 2$ is exponential in gradient descent as well.

While similar gradient descent results are harder to obtain for deeper models, we discuss the general effect of depth on the gradient decent dynamics in appendix C.

## 4    INCREMENTAL LEARNING IN LARGER MODELS

So far, we have only shown interesting properties of incremental learning caused by depth for a toy model. In this section, we will relate several deep models to our toy model and show how incremental learning presents itself in larger models as well.

### 4.1    MATRIX SENSING

The task of matrix sensing is a generalization of matrix completion, where our input space is $\mathcal{X} = \mathbb{R}^{d\times d}$ and our model is a matrix $W \in \mathbb{R}^{d\times d}$, such that:

$$f_W(A) = \langle W, A \rangle = tr\big(W^T A\big)$$

Following Arora et al. (2019), we introduce depth by parameterizing the model using a product of matrices and the following initialization scheme ($W_i \in \mathbb{R}^{d\times d}$):

$$W = W_N W_{N-1} \cdots W_1 \tag{5}$$
$$\forall i \in [N],\ W_i(0) = \sqrt[N]{\sigma_0}I$$

Note that when $d = 1$, the deep matrix sensing model reduces to our toy model without weight sharing. We study the dynamics of the model under gradient flow over a depth-normalized squared loss, assuming the data is labeled by a matrix sensing model parameterized by a PSD $W^* \in \mathbb{R}^{d\times d}$:

$$\begin{aligned}\ell_N(W_N W_{N-1} \cdots W_1) &= \frac{1}{2N}\mathbb{E}_A[(\langle(W_N W_{N-1} \cdots W_1) - W^*, A\rangle^2] \\ &= \frac{1}{2N}||(W_N W_{N-1} \cdots W_1) - W^*||_F^2\end{aligned} \tag{6}$$

The following theorem relates the deep matrix sensing model to our toy model, showing the two have the same dynamical equations:

**Theorem 4.** *Optimizing the deep matrix sensing model described in* (5) *with gradient flow over the depth normalized squared loss ((6)), with weights initialized as in* (5) *leads to the following dynamical equations for different values of* $N$*:*

$$\dot{\sigma}_i(t) = \sigma_i(t)^{2 - \frac{2}{N}}(\sigma_i^* - \sigma_i(t))$$

*Where* $\sigma_i$ *and* $\sigma_i^*$ *are the* $i$th *singular values of* $W$ *and* $W^*$*, respectively, corresponding to the same singular vector.*

The proof follows that of Saxe et al. (2013) and Gidel et al. (2019) and is deferred to appendix E.

Theorem 4 shows us that the bias towards sparse solutions introduced by depth in the toy model is equivalent to the bias for low-rank solutions in the matrix sensing task. This bias was studied in a more general setting in Arora et al. (2019), with empirical results supporting the effect of depth on the obtainment of low-rank solutions under a more natural loss and initialization scheme. We recreate and discuss these experiments and their connection to our analysis in appendix E, and an example of these dynamics in deep matrix sensing can also be seen in panel (a) of figure 1.

## 4.2 QUADRATIC NEURAL NETWORKS

By drawing connections between quadratic networks and matrix sensing (as in Soltanolkotabi et al. (2018)), we can extend our results to these nonlinear models. We will study a simplified quadratic network, where our input space is $\mathcal{X} = \mathbb{R}^d$ and the first layer is parameterized by a weight matrix $W \in \mathbb{R}^{d \times d}$ and followed by a quadratic activation function. The final layer will be a summation layer. We assume, like before, that the labeling function is a quadratic network parameterized by $W^* \in \mathbb{R}^{d \times d}$. Our model can be written in the following way, using the following orthogonal initialization scheme:

$$f_W(x) = \sum_{i=1}^d (w_i^T x)^2 = x^T W^T W x = \langle W^T W, xx^T \rangle \tag{7}$$
$$W_0^T W_0 = \sigma_0 I$$

Immediately, we see the similarity of the quadratic network to the deep matrix sensing model with $N = 2$, where the input space is made up of rank-1 matrices. However, the change in input space forces us to optimize over a different loss function to reproduce the same dynamics:

**Definition 2.** *Given an input distribution over an input space* $\mathcal{X}$ *with a labeling function* $y : \mathcal{X} \to \mathbb{R}$ *and a hypothesis* $h$*, the variance loss is defined in the following way:*

$$\ell_{var}(h) = \frac{1}{16}\mathbb{E}_x[(y(x) - h(x))^2] - \frac{1}{16}\mathbb{E}_x[y(x) - h(x)]^2$$

Note that minimizing this loss function amounts to minimizing the variance of the error, while the squared loss minimizes the second moment of the error. We note that both loss functions have the same minimum for our problem, and the dynamics of the squared loss can be approximated in certain cases by the dynamics of the variance loss. For a complete discussion of the two losses, including the cases where the two losses have similar dynamics, we refer the reader to appendix F.

**Theorem 5.** *Minimizing the quadratic network described and initialized as in* (7) *with gradient flow over the variance loss defined in* (2) *leads to the following dynamical equations:*

$$\dot{\sigma}_i(t) = \sigma_i(t)(\sigma_i^* - \sigma_i(t))$$

*Where* $\sigma_i$ *and* $\sigma_i^*$ *are the* $i$th *singular values of* $W$ *and* $W^*$*, respectively, corresponding to the same singular vector.*

We defer the proof to appendix F and note that these dynamics are the same as our depth-2 toy model, showing that shallow quadratic networks can exhibit incremental learning (albeit requiring a small initialization).

### 4.3 DIAGONAL/CONVOLUTIONAL LINEAR NETWORKS

While incremental learning has been described for deep linear networks in the past, it has been restricted to regression tasks. Here, we illustrate how incremental learning presents itself in binary classification, where implicit bias results have so far focused on convergence at $t \to \infty$ (Soudry et al. (2018), Nacson et al. (2018), Ji & Telgarsky (2019)). Deep linear networks with diagonal weight matrices have been shown to be biased towards sparse solutions when $N > 1$ in Gunasekar et al. (2018), and biased towards the max-margin solution for $N = 1$. Instead of analyzing convergence at $t \to \infty$, we intend to show that the model favors sparse solutions for the entire duration of optimization, and that this is due to the dynamics of incremental learning.

Our theoretical illustration will use our toy model as in (1) (initialized as in (3)) as a special weight-shared case of deep networks with diagonal weight matrices, and we will then show empirical results for the more general setting. We analyze the optimization dynamics of this model over a separable dataset $\{x_i, y_i\}_{i=1}^m$ where $y_i \in \{\pm 1\}$. We use the exponential loss $(\ell(f(x), y) = e^{-yf(x)})$ for the theoretical illustration and experiment on the exponential and logistic losses.

Computing the gradient for the model over $w$, the gradient flow dynamics for $\sigma$ become:

$$\dot{\sigma}_i = \frac{N^2}{m} \sigma_i^{2-\frac{2}{N}} \sum_{j=1}^m e^{-y_j \langle \beta, x_j \rangle} x_j$$

We see the same dynamical attenuation of small values of $\sigma$ that is seen in the regression model, caused by the multiplication by $\sigma_i^{2-\frac{2}{N}}$. From this, we can expect the same type of incremental learning to occur - weights of $\sigma$ will be learned incrementally until the dataset can be separated by the current support of $\sigma$. Then, the dynamics strengthen the growth of the current support while relatively attenuating that of the other values. Since the data is separated, increasing the values of the current support reduces the loss and the magnitude of subsequent gradients, and so we should expect the support to remain the same and the model to converge to a sparse solution.

Granted, the above description is just intuition, but panel (c) of figure 1 shows how it is born out in practice (similar results are obtained for the logistic loss). In appendix G we further explore this model, showing deeper networks have a stronger bias for sparsity. We also observe that the initialization scale plays a similar role as before - deep models are less biased towards sparsity when $\sigma_0$ is large.

In their work, Gunasekar et al. (2018) show an equivalence between the diagonal network and the circular-convolutional network in the frequency domain. According to their results, we should expect to see the same sparsity-bias of diagonal networks in convolutional networks, when looking at the Fourier coefficients of $\sigma$. An example of this can be seen in panel (d) of figure 1, and we refer the reader to appendix G for a full discussion of their convolutional model and it's incremental learning dynamics.

## 5 CONCLUSION

Gradient-based optimization for deep linear models has an implicit bias towards simple (sparse) solutions, caused by an incremental search strategy over the hypothesis space. Deeper models have a stronger tendency for incremental learning, exhibiting it in more realistic initialization scales.

This dynamical phenomenon exists for the entire optimization process for regression as well as classification tasks, and for many types of models - diagonal networks, convolutional networks, matrix completion and even the nonlinear quadratic network. We believe this kind of dynamical analysis may be able to shed light on the generalization of deeper nonlinear neural networks as well, with shallow quadratic networks being only a first step towards that goal.

ACKNOWLEDGMENTS

This research is supported by the European Research Council (TheoryDL project).

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

## A   PROOF OF THEOREM 2

**Theorem.** *Given two values $\sigma_i, \sigma_j$ of a toy linear model as in (1), such that $\frac{\sigma_i^*}{\sigma_j^*} = r > 1$ and the model is initialized as in (3), and given two scalars $s \in (0, \frac{1}{4})$ and $f \in (\frac{3}{4}, 1)$, then the largest initialization value for which the learning phases of the values are $(s, f)$-incremental, denoted $\sigma_0^{th}$, is lower and upper bounded in the following way:*

$$s\sigma_j^* \left( \frac{s}{rf} \right)^{\frac{1}{(1-f)(r-1)}} \leq \sigma_0^{th} \leq s\sigma_j^* \left( \frac{s}{rf} \right)^{\frac{1-s}{r-1}} \qquad\qquad N = 2$$

$$s\sigma_j^* \left( \frac{(1-f)(r-1)}{1+(1-f)(r-1)} \right)^{\frac{N}{N-2}} \leq \sigma_0^{th} \leq s\sigma_j^* \left( \frac{r-1}{r-s} \right)^{\frac{N}{N-2}} \qquad\qquad N \geq 3$$

*Proof.* Our strategy will be to define the time $t_\alpha$ for which a value reaches a fraction $\alpha$ of it's optimal value, and then require that $t_f(\sigma_i) \leq t_s(\sigma_j)$. We begin with recalling the differential equation which determines the dynamics of the model:

$$\dot{\sigma} = \sigma^{2-\frac{2}{N}} (\sigma^* - \sigma)$$

Since the solution for $N \geq 3$ is implicit and difficult to manage in a general form, we will define $t_\alpha$ using the integral of the differential equation. The equation is separable, and under initialization of $\sigma_0$ we can describe $t_\alpha(\sigma)$ in the following way:

$$t_\alpha(\sigma) = \int_{\sigma_0}^{\alpha\sigma^*} \frac{d\sigma}{\sigma^{2-\frac{2}{N}}(\sigma^* - \sigma)}$$

Incremental learning takes place when $\sigma_i(t_f) = f\sigma_i^*$ happens before $\sigma_j(t_s) = s\sigma_j^*$. We can write this condition in the following way:

$$\int_{\sigma_0}^{f\sigma_i^*} \frac{d\sigma}{\sigma^{2-\frac{2}{N}}(\sigma_i^* - \sigma)} \leq \int_{\sigma_0}^{s\sigma_j^*} \frac{d\sigma}{\sigma^{2-\frac{2}{N}}(\sigma_j^* - \sigma)}$$

Plugging in $\sigma_i = r\sigma_j$ and rearranging, we get the following necessary and sufficient condition for incremental learning:

$$\int_{\sigma_0}^{fr\sigma_j^*} \frac{d\sigma}{\sigma^{2-\frac{2}{N}}(1 - \frac{\sigma}{r\sigma_j^*})} \leq r \int_{\sigma_0}^{s\sigma_j^*} \frac{d\sigma}{\sigma^{2-\frac{2}{N}}(1 - \frac{\sigma}{\sigma_j^*})}$$

Our last step before relaxing and restricting our condition will be to split the integral on the left-hand side into two integrals:

$$\int_{\sigma_0}^{s\sigma_j^*} \frac{d\sigma}{\sigma^{2-\frac{2}{N}}(1 - \frac{\sigma}{r\sigma_j^*})} + \int_{s\sigma_j^*}^{fr\sigma_j^*} \frac{d\sigma}{\sigma^{2-\frac{2}{N}}(1 - \frac{\sigma}{r\sigma_j^*})} \leq r \int_{\sigma_0}^{s\sigma_j^*} \frac{d\sigma}{\sigma^{2-\frac{2}{N}}(1 - \frac{\sigma}{\sigma_j^*})} \qquad (8)$$

At this point, we cannot solve this equation and isolate $\sigma_0$ to obtain a clear threshold condition on it for incremental learning. Instead, we will relax/restrict the above condition to get a necessary/sufficient condition on $\sigma_0$, leading to a lower and upper bound on the threshold value of $\sigma_0$.

SUFFICIENT CONDITION

To obtain a sufficient (but not necessary) condition on $\sigma_0$, we may make the condition stricter either by increasing the left-hand side or decreasing the right-hand side. We can increase the left-hand side by removing $r$ from the left-most integral's denominator ($r > 1$) and then combine the left-most and right-most integrals:

$$\int_{s\sigma_j^*}^{fr\sigma_j^*} \frac{d\sigma}{\sigma^{2-\frac{2}{N}}\left(1 - \frac{\sigma}{r\sigma_j^*}\right)} \leq (r-1)\int_{\sigma_0}^{s\sigma_j^*} \frac{d\sigma}{\sigma^{2-\frac{2}{N}}\left(1 - \frac{\sigma}{\sigma_j^*}\right)}$$

Next, we note that the integration bounds give us a bound on $\sigma$ for either integral. This means we can replace $1 - \frac{\sigma}{\sigma_j^*}$ with $1$ on the right-hand side, and replace $1 - \frac{\sigma}{r\sigma_j^*}$ with $1 - f$ on the left-hand side:

$$\frac{1}{1-f}\int_{s\sigma_j^*}^{fr\sigma_j^*} \frac{d\sigma}{\sigma^{2-\frac{2}{N}}} \leq (r-1)\int_{\sigma_0}^{s\sigma_j^*} \frac{d\sigma}{\sigma^{2-\frac{2}{N}}}$$

We may now solve these integrals for every $N$ and isolate $\sigma_0$, obtaining the lower bound on $\sigma_0^{th}$. We start with the case where $N = 2$:

$$\frac{1}{1-f}\left(\log(fr\sigma_j^*) - \log(s\sigma_j^*)\right) \leq (r-1)\left(\log(s\sigma_j^*) - \log(\sigma_0)\right)$$

Rearranging to isolate $\sigma_0$, we obtain our result:

$$\sigma_0 \leq s\sigma_j^*\left(\frac{s}{rf}\right)^{\frac{1}{(1-f)(r-1)}}$$

For the $N \geq 3$ case, we have the following after solving the integrals:

$$\frac{1}{1-f}\left(\left(\frac{1}{s\sigma_j^*}\right)^{1-\frac{2}{N}} - \left(\frac{1}{rf\sigma_j^*}\right)^{1-\frac{2}{N}}\right) \leq (r-1)\left(\left(\frac{1}{\sigma_0}\right)^{1-\frac{2}{N}} - \left(\frac{1}{s\sigma_j^*}\right)^{1-\frac{2}{N}}\right)$$

For simplicity we may further restrict the condition by removing the term $\left(\frac{1}{rf\sigma_j^*}\right)^{1-\frac{2}{N}}$. Solving for $\sigma_0$ gives us the following:

$$\sigma_0 \leq s\sigma_j^*\left(\frac{(1-f)(r-1)}{1+(1-f)(r-1)}\right)^{\frac{N}{N-2}}$$

NECESSARY CONDITION

To obtain a necessary (but not sufficient) condition on $\sigma_0$, we may relax the condition in (8) either by decreasing the left-hand side or increasing the right-hand side. We begin by rearranging the equation:

$$\int_{s\sigma_j^*}^{fr\sigma_j^*} \frac{d\sigma}{\sigma^{2-\frac{2}{N}}\left(1 - \frac{\sigma}{r\sigma_j^*}\right)} \leq r\int_{\sigma_0}^{s\sigma_j^*} \frac{d\sigma}{\sigma^{2-\frac{2}{N}}\left(1 - \frac{\sigma}{\sigma_j^*}\right)} - \int_{\sigma_0}^{s\sigma_j^*} \frac{d\sigma}{\sigma^{2-\frac{2}{N}}\left(1 - \frac{\sigma}{r\sigma_j^*}\right)}$$

Like before, we may use the integration bounds to bound $\sigma$. Plugging in $\sigma = s\sigma_j^*$ for all integrals decreases the left-hand side and increases the right-hand side, leading us to the following:

$$\frac{r}{r-s}\int_{s\sigma_j^*}^{fr\sigma_j^*} \frac{d\sigma}{\sigma^{2-\frac{2}{N}}} \leq \left(\frac{r}{1-s} - \frac{r}{r-s}\right)\int_{\sigma_0}^{s\sigma_j^*} \frac{d\sigma}{\sigma^{2-\frac{2}{N}}}$$

Rearranging, we get the following inequality:

$$\int_{s\sigma_j^*}^{fr\sigma_j^*} \frac{d\sigma}{\sigma^{2-\frac{2}{N}}} \leq \frac{r-1}{1-s} \int_{\sigma_0}^{s\sigma_j^*} \frac{d\sigma}{\sigma^{2-\frac{2}{N}}}$$

We now solve the integrals for the different cases. For $N = 2$, we have:

$$\log(fr\sigma_j^*) - \log(s\sigma_j^*) \leq \frac{r-1}{1-s}\Big( \log(s\sigma_j^*) - \log(\sigma_0) \Big)$$

Rearranging to isolate $\sigma_0$, we get our condition:

$$\sigma_0 \leq s\sigma_j^* \Big(\frac{s}{rf}\Big)^{\frac{1-s}{r-1}}$$

Finally, for $N \geq 3$, we solve the integrals to give us:

$$\Big( \big(\frac{1}{s\sigma_j^*}\big)^{1-\frac{2}{N}} - \big(\frac{1}{rf\sigma_j^*}\big)^{1-\frac{2}{N}} \Big) \leq \frac{r-1}{1-s} \Big( \big(\frac{1}{\sigma_0}\big)^{1-\frac{2}{N}} - \big(\frac{1}{s\sigma_j^*}\big)^{1-\frac{2}{N}} \Big)$$

Rearranging to isolate $\sigma_0$, we get our condition:

$$\sigma_0 \leq s\sigma_j^* \Big(\frac{r-1}{r-s}\Big)^{\frac{N}{N-2}}$$

SUMMARY

For a given $N$, we derived a sufficient condition and a necessary condition on $\sigma_0$ for $(s, f)$-incremental learning. The necessary and sufficient condition on $\sigma_0$, which is the largest initialization value for which we see incremental learning (denoted $\sigma_0^{th}$), is between the two derived bounds.

The precise bounds can possibly be improved a bit, but the asymptotic dependence on $r$ is the crux of the matter, showing the dependence on $r$ changes with depth with a substantial difference when we move from shallow models ($N = 2$) to deeper ones ($N \geq 3$)

$\square$

## B  PROOF OF THEOREM 3

**Theorem.** *Given two values $\sigma_i, \sigma_j$ of a depth-2 toy linear model as in (1), such that $\frac{\sigma_i^*}{\sigma_j^*} = r > 1$ and the model is initialized as in (3), and given two scalars $s \in (0, \frac{1}{4})$ and $f \in (\frac{3}{4}, 1)$, and assuming $\sigma_j^* \geq 2\sigma_0$, and assuming we optimize with gradient descent with a learning rate $\eta \leq \frac{c}{\sigma_1^*}$ for $c < 2(\sqrt{2} - 1)$ and $\sigma_1^*$ the largest value of $\sigma^*$, then the largest initialization value for which the learning phases of the values are $(s, f)$-incremental, denoted $\sigma_0^{th}$, is lower and upper bounded in the following way:*

$$\frac{1}{2}\frac{s}{1-s}\sigma_j^*\Big(\frac{1-f}{2rf}\frac{s}{1-s}\Big)^{\frac{1}{A-1}} \leq \sigma_0^{th} \leq \frac{s}{1-s}\sigma_j^*\Big(\frac{1-f}{f}\frac{s}{1-s}\Big)^{\frac{1}{B-1}}$$

*Where A and B are defined as:*

$$A = \frac{\log\big(1 - c\frac{\sigma_i^*}{\sigma_1^*} + c^2\big(\frac{\sigma_i^*}{\sigma_1^*}\big)^2\big)}{\log\big(1 - c\frac{\sigma_j^*}{\sigma_1^*} - \frac{c^2}{4}\big(\frac{\sigma_j^*}{\sigma_1^*}\big)^2\big)} \qquad B = \frac{\log\big(1 - c\frac{\sigma_i^*}{\sigma_1^*} - \frac{c^2}{4}\big(\frac{\sigma_i^*}{\sigma_1^*}\big)^2\big)}{\log\big(1 - c\frac{\sigma_j^*}{\sigma_1^*} + c^2\big(\frac{\sigma_j^*}{\sigma_1^*}\big)^2\big)}$$

*Proof.* To show our result for gradient descent and $N = 2$, we build on the proof techniques of theorem 3 of Gidel et al. (2019). We start by deriving the recurrence relation for the values $\sigma(t)$ for general depth, when $t$ now stands for the iteration. Remembering that $w_i^n = \sigma_i$, we write down the gradient update for $w_i(t)$:

$$w_i(t+1) = w_i(t) + \eta \frac{1}{N} w_i(t)^{N-1}(\sigma_i^* - \sigma_i(t)) = \sqrt[N]{\sigma_i(t)} + \eta \frac{1}{N} \sigma_i(t)^{1-\frac{1}{N}}(\sigma_i^* - \sigma_i(t))$$

Raising $w_i(t)$ to the $N$th power, we get the gradient update for the $\sigma$ values:

$$\sigma_i(t+1) = \left( \sqrt[N]{\sigma_i(t)} + \eta \frac{1}{N}(\sigma_i^* - \sigma_i(t))\sigma_i(t)^{1-\frac{1}{N}} \right)^N = \sigma_i(t)\left( 1 + \eta \frac{1}{N}\sigma_i(t)^{1-\frac{2}{N}}(\sigma_i^* - \sigma_i(t)) \right)^N \tag{9}$$

Next, we will prove a simple lemma which gives us the maximal learning rate we will consider for the analysis, for which there is no overshooting (the values don't grow larger than the optimal values).

**Lemma 1.** *For the gradient update in* (9)*, assuming $\sigma_i(0) < \sigma_i^*$, if $\eta \leq \left( \frac{1}{\sigma_1^*} \right)^{2-\frac{2}{N}}$ and $N \geq 2$, then:*

$$\forall t : \sigma_i(t) \leq \sigma_i^*$$

*Proof.* Plugging in $\eta = c \left( \frac{1}{\sigma_i^*} \right)^{2-\frac{2}{N}}$ for $c \leq 1$, we have:

$$\sigma_i(t+1) = \sigma_i(t)\left( 1 + \frac{c}{N}\left( \frac{\sigma_i(t)}{\sigma_i^*} \right)^{1-\frac{2}{N}}\left( 1 - \left( \frac{\sigma_i(t)}{\sigma_i^*} \right) \right) \right)^N \leq \sigma_i(t)e^{\left( \frac{\sigma_i(t)}{\sigma_i^*} \right)^{1-\frac{2}{N}}\left( 1 - \left( \frac{\sigma_i(t)}{\sigma_i^*} \right) \right)}$$

Defining $r_i = \frac{\sigma_i}{\sigma_i^*}$ and dividing both sides by $\sigma_i^*$, we have:

$$r_i(t+1) \leq r_i(t)e^{r_i(t)^{1-\frac{2}{N}}(1-r_i(t))}$$

It is enough to show that for any $0 \leq r \leq 1$, we have that $re^{r^{1-\frac{2}{N}}(1-r)} \leq 1$, as over-shooting occurs when $r_i(t) > 1$. Indeed, this function is monotonic increasing in $0 \leq r \leq 1$ (since the exponent is non negative), and equals 1 when $r = 1$. Since $r = 1$ is a fixed point and no iteration that starts at $r < 1$ can cross 1, then $r_i(t) \leq 1$ for any $t$. This concludes our proof. □

Under this choice of learning rate, we can now obtain our incremental learning results for gradient descent when $N = 2$. Our strategy will be bounding $\sigma_i(t)$ from below and above, which will give us a lower and upper bound for $t_\alpha(\sigma_i)$. Once we have these bounds, we will be able to describe either a necessary or a sufficient condition on $\sigma_0$ for incremental learning, similar to theorem 2.

The update rule for $N = 2$ is:

$$\sigma_i(t+1) = \sigma_i(t)\left( 1 + \frac{1}{2}\eta(\sigma_i^* - \sigma_i(t)) \right)^2$$

Next, we plug in $\eta = \frac{c}{\sigma_1^*}$ for $c < 2(\sqrt{2} - 1) < 1$ and denote $R_i = \frac{\sigma_i^*}{\sigma_1^*} \leq 1$ and $r_i(t) = \frac{\sigma_i(t)}{\sigma_i^*} \leq 1$ to get:

$$\sigma_i(t+1) = \sigma_i(t)\left( 1 + \frac{c}{2}R_i(1 - r_i(t)) \right)^2$$

Following theorem 3 of Gidel et al. (2019), we bound $\frac{1}{\sigma_i(t)}$:

$$
\begin{aligned}
\frac{1}{\sigma_i(t+1)} &= \frac{1}{\sigma_i(t)} \frac{1}{\left(1 + \frac{c}{2}R_i(1 - r_i(t))\right)^2} \\
&= \frac{1}{\sigma_i(t)} \frac{1}{1 + (1 - r_i(t))\left(cR_i + \frac{c^2}{4}R_i^2(1 - r_i(t))\right)} \\
&\geq \frac{1}{\sigma_i(t)} \frac{1}{1 + (1 - r_i(t))\left(cR_i + \frac{c^2}{4}R_i^2\right)} \\
&\geq \frac{1}{\sigma_i(t)} \left(1 - (1 - r_i(t))(cR_i + \frac{c^2}{4}R_i^2)\right) \\
&= \frac{1}{\sigma_i(t)} - \left(\frac{1}{\sigma_i(t)} - \frac{1}{\sigma_i^*}\right)(cR_i + \frac{c^2}{4}R_i^2)
\end{aligned}
$$

Where in the fourth line we use the inequality $\frac{1}{1+x} \geq 1 - x, \ \forall x \geq 0$. We may now subtract $\frac{1}{\sigma_i^*}$ from both sides to obtain:

$$
\frac{1}{\sigma_i(t)} - \frac{1}{\sigma_i^*} \geq \left(\frac{1}{\sigma_i(t-1)} - \frac{1}{\sigma_i^*}\right)\left(1 - cR_i - \frac{c^2}{4}R_i^2\right) \geq \left(\frac{1}{\sigma_0} - \frac{1}{\sigma_i^*}\right)\left(1 - cR_i - \frac{c^2}{4}R_i^2\right)^t
$$

We may now obtain a bound on $t_\alpha(\sigma_i)$ by plugging in $\sigma_i(t) = \alpha\sigma_i^*$ and taking the log:

$$
\log\left(\frac{1 - \alpha}{\alpha\left(\frac{\sigma_i^*}{\sigma_0} - 1\right)}\right) \geq t_\alpha \cdot \log\left(1 - cR_i - \frac{c^2}{4}R_i^2\right)
$$

Rearranging (note that $\log\left(1 - cR_i - \frac{c^2}{4}R_i^2\right) < 0$ and that our choice of $c$ keeps the argument of the log positive), we get:

$$
t_\alpha(\sigma_i) \geq \frac{\log\left(\frac{1-\alpha}{\alpha\left(\frac{\sigma_i^*}{\sigma_0} - 1\right)}\right)}{\log\left(1 - cR_i - \frac{c^2}{4}R_i^2\right)}
$$

Next, we follow the same procedure for an upper bound. Starting with our update step:

$$
\begin{aligned}
\frac{1}{\sigma_i(t+1)} &= \frac{1}{\sigma_i(t)} \frac{1}{1 + (1 - r_i(t))\left(cR_i + \frac{c^2}{4}R_i^2(1 - r_i(t))\right)} \\
&\leq \frac{1}{\sigma_i(t)} \frac{1}{1 + (1 - r_i(t))cR_i} \\
&\leq \frac{1}{\sigma_i(t)} \left(1 - (1 - r_i(t))cR_i + (1 - r_i(t))^2 c^2 R_i^2\right)
\end{aligned}
$$

Where in the last line we use the inequality $\frac{1}{1+x} \leq 1 - x + x^2, \ \forall x \geq 0$. Subtracting $\frac{1}{\sigma_i^*}$ from both sides, we get:

$$
\frac{1}{\sigma_i(t)} - \frac{1}{\sigma_i^*} \leq \left(\frac{1}{\sigma_i(t-1)} - \frac{1}{\sigma_i^*}\right)\left(1 - cR_i + c^2 R_i^2\right) \leq \left(\frac{1}{\sigma_0} - \frac{1}{\sigma_i^*}\right)\left(1 - cR_i + c^2 R_i^2\right)^t
$$

Rearranging like before, we get the bound on the $\alpha$-time:

$$
t_\alpha(\sigma_i) \leq \frac{\log\left(\frac{1-\alpha}{\alpha\left(\frac{\sigma_i^*}{\sigma_0} - 1\right)}\right)}{\log\left(1 - cR_i + c^2 R_i^2\right)}
$$

Given these bounds, we would like to find the conditions on $\sigma_0$ that allows for $(s, f)$-incremental learning. We will find a sufficient condition and a necessary condition, like in the proof of theorem 2.

SUFFICIENT CONDITION

A sufficient condition for incremental learning will be one which is possibly stricter than the exact condition. We can find such a condition by requiring the upper bound of $t_f(\sigma_i)$ to be smaller than the lower bound on $t_s(\sigma_j)$. This becomes the following condition:

$$\frac{\log\left(\frac{1-f}{f}\frac{\sigma_0}{\sigma_i^* - \sigma_0}\right)}{\log\left(1 - cR_i + c^2 R_i^2\right)} \leq \frac{\log\left(\frac{1-s}{s}\frac{\sigma_0}{\sigma_j^* - \sigma_0}\right)}{\log\left(1 - cR_j - \frac{c^2}{4}R_j^2\right)}$$

Defining $A = \frac{\log\left(1 - cR_i + c^2 R_i^2\right)}{\log\left(1 - cR_j - \frac{c^2}{4}R_j^2\right)}$ and rearranging, we get the following:

$$\log\left(\frac{1-f}{f}\frac{\sigma_0}{\sigma_i^* - \sigma_0}\right) \geq A\log\left(\frac{1-s}{s}\frac{\sigma_0}{\sigma_j^* - \sigma_0}\right)$$

We may now take the exponent of both sides and rearrange again, remembering $\frac{\sigma_i^*}{\sigma_j^*} = r > 1$, to get the following condition:

$$\frac{(\sigma_j^* - \sigma_0)^A}{\sigma_0^{A-1}(r\sigma_j^* - \sigma_0)} \geq \frac{f}{1-f}\left(\frac{1-s}{s}\right)^A$$

Now, we will add the very reasonable assumption that $\sigma_j^* \geq 2\sigma_0$, which allows us to replace $\frac{\sigma_j^* - \sigma_0}{r\sigma_j^* - \sigma_0}$ with $\frac{1}{2r}$ and replace $(\sigma_j^* - \sigma_0)^{A-1}$ with $(\frac{1}{2}\sigma_j)^{A-1}$, only making the condition stricter. This simplifies the expression to the following:

$$\left(\frac{\sigma_j^*}{\sigma_0}\right)^{A-1} \geq \frac{rf}{1-f}\left(\frac{2-2s}{s}\right)^A$$

Now we can rearrange and isolate $\sigma_0$ to get a sufficient condition for incremental learning:

$$\sigma_0 \leq \frac{1}{2}\frac{s}{1-s}\sigma_j^*\left(\frac{1-f}{2rf}\frac{s}{1-s}\right)^{\frac{1}{A-1}}$$

NECESSARY CONDITION

A necessary condition for incremental learning will be one which is possibly more relaxed than the exact condition. We can find such a condition by requiring the lower bound of $t_f(\sigma_i)$ to be smaller than the upper bound on $t_s(\sigma_j)$. This becomes the following condition:

$$\frac{\log\left(\frac{1-f}{f}\frac{\sigma_0}{\sigma_i^* - \sigma_0}\right)}{\log\left(1 - cR_i - \frac{c^2}{4}R_i^2\right)} \leq \frac{\log\left(\frac{1-s}{s}\frac{\sigma_0}{\sigma_j^* - \sigma_0}\right)}{\log\left(1 - cR_j + c^2 R_j^2\right)}$$

Defining $B = \frac{\log\left(1 - cR_i - \frac{c^2}{4}R_i^2\right)}{\log\left(1 - cR_j + c^2 R_j^2\right)}$ and rearranging, we get the following:

$$\log\left(\frac{1-f}{f}\frac{\sigma_0}{\sigma_i^* - \sigma_0}\right) \geq B\log\left(\frac{1-s}{s}\frac{\sigma_0}{\sigma_j^* - \sigma_0}\right)$$

We may now take the exponent of both sides and rearrange again, remembering $\frac{\sigma_i^*}{\sigma_j^*} = r > 1$, to get the following condition:

$$\frac{(\sigma_j^* - \sigma_0)^B}{\sigma_0^{B-1}(r\sigma_j^* - \sigma_0)} \geq \frac{f}{1-f}\left(\frac{1-s}{s}\right)^B$$

We may now relax the condition further, by removing the $r$ from the denominator of the left-hand side and the $\sigma_0$ from the numerator. This gives us the following:

$$\left(\frac{\sigma_j^*}{\sigma_0}\right)^{B-1} \geq \frac{f}{1-f}\left(\frac{1-s}{s}\right)^B$$

Finally, rearranging gives us the necessary condition:

$$\sigma_0 \leq \frac{s}{1-s}\sigma_j^*\left(\frac{1-f}{f}\frac{s}{1-s}\right)^{\frac{1}{B-1}}$$

$\square$

## C  DISCUSSION OF GRADIENT DESCENT FOR GENERAL $N$

While we were able to generalize our result to gradient descent for $N = 2$, our proof technique relies on the ability to get a non-implicit solution for $\sigma(t)$ which we discretized and bounded. This is harder to generalize to larger values of $N$, where the solution is implicit. Still, we can informally illustrate the effect of depth on the dynamics of gradient descent by approximating the update rule of the values.

We start by reminding ourselves of the gradient descent update rule for $\sigma$, for a learning rate $\eta = c\left(\frac{1}{\sigma_1^*}\right)^{2-\frac{2}{N}}$:

$$\sigma_i(t+1) = \sigma_i(t)\left(1 + \frac{c}{N}\left(\frac{1}{\sigma_1^*}\right)^{2-\frac{2}{N}}\sigma_i(t)^{1-\frac{2}{N}}(\sigma_i^* - \sigma_i(t))\right)^N$$

To compare two values in the same scales, we will divide both sides by the optimal value $\sigma_i^*$ and look at the update step of the ratio $r_i = \frac{\sigma_i(t)}{\sigma_i^*}$, also denoting $R_i = \frac{\sigma_i^*}{\sigma_1^*}$:

$$r_i(t+1) = r_i(t)\left(1 + \frac{c}{N}R_i^{2-\frac{2}{N}}r_i(t)^{1-\frac{2}{N}}(1 - r_i(t))\right)^N$$

We will focus on the early stages of the optimization process, where $r \ll 1$. This means we can neglect the $1 - r_i(t)$ term in the update step, giving us the approximate update step we will use to compare the general $i, j$ values:

$$r_i(t+1) \approx r_i(t)\left(1 + \frac{c}{N}R_i^{2-\frac{2}{N}}r_i(t)^{1-\frac{2}{N}}\right)^N$$

We would like to compare the dynamics of $r_i$ and $r_j$, which is difficult to do when the recurrence relation isn't solvable. However, we can observe the first iteration of gradient descent and see how depth affects this iteration. Since we are dealing with variables which are ratios of different optimal values, the initial values of $r$ are different. Denoting $\mathbf{r} = \frac{\sigma_i^*}{\sigma_j^*}$, we can describe the initialization of $r_j$ using that of $r_i$:

$$r_j(0) = \mathbf{r}r_i(0)$$

Plugging in the initial conditions and noting that $R_i = \mathbf{r}R_j$, we get:

$$r_i(1) \approx r_i(0)\Big(1 + \frac{cR_i^{2-\frac{2}{N}}}{N}r_i(0)^{1-\frac{2}{N}}\Big)^N$$

$$r_j(1) \approx r_i(0)\Big(\sqrt[N]{\mathbf{r}} + \Big(\frac{1}{\mathbf{r}}\Big)^{\frac{N-1}{N}}\frac{cR_i^{2-\frac{2}{N}}}{N}r_i(0)^{1-\frac{2}{N}}\Big)^N$$

We see that the two ratios have a similar update, with the ratio of optimal values playing a role in how large the initial value is versus how large the added value is. When we use a small learning rate, we have a very small $c$ which means we can make a final approximation and neglect the higher order terms of $c$:

$$r_i(1) \approx r_i(0) + cR_i^{2-\frac{2}{N}}r_i(0)^{2-\frac{2}{N}}$$

$$r_j(1) \approx \mathbf{r}r_i(0) + \Big(\frac{1}{\mathbf{r}}\Big)^{\frac{N-1}{N}}cR_i^{2-\frac{2}{N}}r_i(0)^{2-\frac{2}{N}}$$

We can see that while the initial conditions favor $r_j$, the size of the update for $r_i$ is larger by a factor of $r^{\frac{N-1}{N}}$ when the initialization and learning rates are small. This accumulates throughout the optimization, making $r_i$ eventually converge faster than $r_j$.

The effect of depth here is clear - the deeper the model, the larger the relative step size of $r_i$ and the faster it converges relative to $r_j$.

## D    COMPARISON OF THE TOY MODEL AND OMP

Learning our toy model, when it's incremental learning is taken to the limit, can be described as an iterative procedure where at every step an additional feature is introduced such that it's weight is non-zero and then the model is optimized over the current set of features. This description is also relevant for the sparse approximation algorithm orthogonal matching pursuit (Pati et al., 1993), where the next feature is greedily chosen to be the one which most improves the current model.

While the toy model and OMP are very different algorithms for learning sparse linear models, we will show empirically that they behave similarly. This allows us to view incremental learning as a continuous-time extension of a greedy iterative algorithm.

To allow for negative weights in our experiments, we augment our toy model as in the toy model of Woodworth et al. (2019). Our model will have the same induced form as before:

$$f_\sigma(x) = \langle \sigma, x \rangle$$

However, we parameterize $\sigma$ using $w_+, w_- \in \mathbb{R}^d$ in the following way:

$$\sigma_i = w_{+,i}^N - w_{-,i}^N$$
$$\forall i, \ w_{+,i}(0) = w_{-,i}(0) = \sqrt[N]{\sigma_0}$$

We can now treat this algorithm as a sparse approximation pursuit algorithm - given a dictionary $D \in \mathbb{R}^{d \times n}$ and an example $x \in \mathbb{R}^d$, we wish to find the sparsest $\alpha$ for which $D\alpha \approx x$ by minimizing the $\ell_0$ norm of $\alpha$ subject to $||D\alpha - x||_2^2 = 0$[3]. Under this setting, we can compare OMP to our toy model by comparing the sets of features that the two algorithms choose for a given example and dictionary.

In figure 3 we run such a comparison. Using a dictionary of 1000 atoms and an example of dimensionality 80 sampled from a random hidden vector of a given sparsity $s$, we run both algorithms and record the first $s$ features chosen[4].

---

[3]Note that this problem is equivalent to learning the toy model over the squared loss, where the examples in the original learning problem play the role of the dictionary.

[4]For the toy model, we define a feature to be "chosen" once it's $\sigma_i$ value passes some small threshold in absolute value.

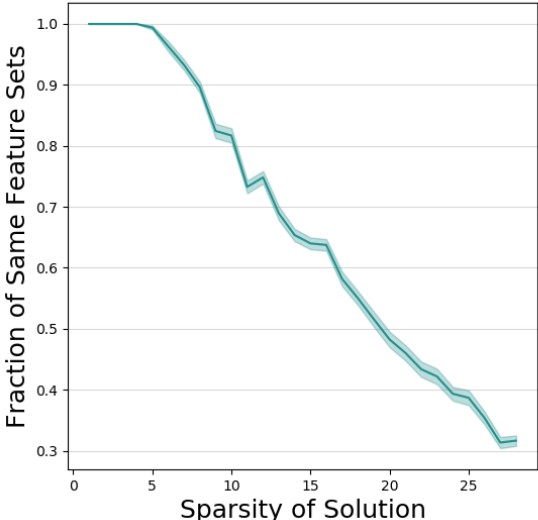

Figure 3: Empirical comparison of the dynamics of the toy model to OMP. The toy model has a depth of 5 and was initialized with a scale of 1e-4 and a learning rate of 3e-3. We compare the fraction of agreement between the sets of first $s$ features selected of the two algorithms for every given sparsity level $s$, averaged over 100 experiments (the shaded regions are empirical standard deviations). For example, for sparsity level 3, we look at the sets of first 3 features selected by each algorithm and calculate the fraction of them that appear in both sets.

For every sparsity $s$, we plot the mean fraction of agreement between the sets of features chosen by OMP and the toy model over 100 experiments. We see that the two algorithms choose very similar features at the beginning, suggesting that the deep model approximates the discrete behavior of OMP. Only when the number of features increases do we see that the behavior of the two models begins to differ, caused by the fact that the toy model has a finite initialization scale and learning rate.

These experiments demonstrate the similarity between the incremental learning of deep models and the discrete behavior of greedy approximation algorithms such as OMP. Adopting this view also allows us to put our finger on another strength of the dynamics of deep models - while greedy algorithms such as OMP require the analytical solution or approximation of every iterate, the dynamics of deep models are able to incrementally learn any differentiable function. For example, looking back at the matrix sensing task and the classification models in section 4, we see that while there isn't an immediate and efficient extension of OMP for these settings, the dynamics of learning deep models extends naturally and exhibits the same incremental learning as OMP.

# E    INCREMENTAL LEARNING IN MATRIX SENSING

## E.1    PROOF OF THEOREM 4

**Theorem.** *Minimizing the deep matrix sensing model described in* (5) *with gradient flow over the depth normalized squared loss* (6)*, with Gaussian inputs and weights initialized as in* (5) *leads to the following dynamical equations for different values of $N$:*

$$\dot{\sigma}_i(t) = \sigma_i(t)^{2-\frac{2}{N}}(\sigma_i^* - \sigma_i(t))$$

*Where $\sigma_i$ and $\sigma_i^*$ are the $i$th singular values of $W$ and $W^*$, respectively, corresponding to the same singular vectors.*

*Proof.* We will adapt the proof from Saxe et al. (2013) for multilayer linear networks. The gradient flow equations for $W_n$, $n \in [N]$ are:

$$\dot{W}_n = \frac{1}{N} W_{1:n-1}^T (W^* - W) W_{n+1:N}^T$$

Where we denote $W_{j:k} = \prod_{i=j}^{k} W_i$.

Since we assumed $W^*$ is (symmetric) PSD, there exists an orthogonal matrix $U$ for which $D^* = UW^*U^T$ where $D^*$ is diagonal. Under the initialization in (3), $U$ diagonalizes all $W_n$ matrices at initialization such that $D_n = UW_nU^T = \sqrt[N]{\sigma_0} I$. Making this change of variables for all $W_n$, we get:

$$\dot{W}_n = \frac{1}{N} U^T D_{1:n-1} U (W^* - W) U^T D_{n+1:N} U$$

Rearranging, we get a set of decoupled differential equations for the singular values of $W_n$:

$$\dot{D}_n = \frac{1}{N} D_{1:n-1} (D^* - D) D_{n+1:N}$$

Note that since these matrices are all diagonal at initialization, the above dynamics ensure that they remain diagonal throughout the optimization. Denoting $\sigma_{n,i}$ as the $i$'th singular value of $W_n$ and $\sigma_i$ as the $i$'th singular value of $W$, we get the following differential equation:

$$\dot{\sigma}_{n,i} = \frac{1}{N} (\sigma_i^* - \sigma_i) \prod_{j \neq n} \sigma_{j,i}$$

Since we assume at initialization that $\forall n, m, i : \sigma_{n,i}(0) = \sigma_{m,i}(0) = \sqrt[N]{\sigma_0}$, the above dynamics are the same for all singular values and we get $\forall n, m, i : \sigma_{n,i}(t) = \sigma_{m,i}(t) = \sqrt[N]{\sigma_i(t)}$. We may now use this to calculate the dynamics of the singular value of $W$, since they are the product the the singular values of all $W_n$ matrices. Denoting $\sigma_{-n,i} = \prod_{k \neq n} \sigma_{k,i}$ and using the chain rule:

$$\dot{\sigma}_i = \sum_{n=1}^{N} \sigma_{-n,i} \dot{\sigma}_{n,i} = \sigma_i^{2-\frac{2}{N}} (\sigma_i^* - \sigma_i)$$

$\square$

### E.2 EMPIRICAL EXAMINATION

Our analytical results are only applicable for the population loss over Gaussian inputs. These conditions are far from the ones used in practice and studied in Arora et al. (2019), where the problem is over-determined and the weights are drawn from a Gaussian distribution with a small variance. To show our conclusions regarding incremental learning extend qualitatively to more natural settings, we empirically examine the deep matrix sensing model in this natural setting for different depths and initialization scales as seen in figure 4.

Notice how incremental learning is exhibited even when the number of examples is much smaller than the number of parameters in the model. While we can't rely on our theory for describing the exact dynamics of the optimization for these kinds of over-determined problems, the qualitative conclusions we get from it are still applicable.

Another interesting phenomena we should note is that once the dataset becomes very small (the second row of the figure), we see all "currently active" singular values change at the beginning of every new phase (this is best seen in the bottom-right panel). This suggests that since there is more than one optimal solution, once we increase the current rank of our model it may find a solution that has a different set of singular values and vectors and thus all singular values change at the beginning of a new learning phase. This demonstrates the importance of incremental learning for obtaining

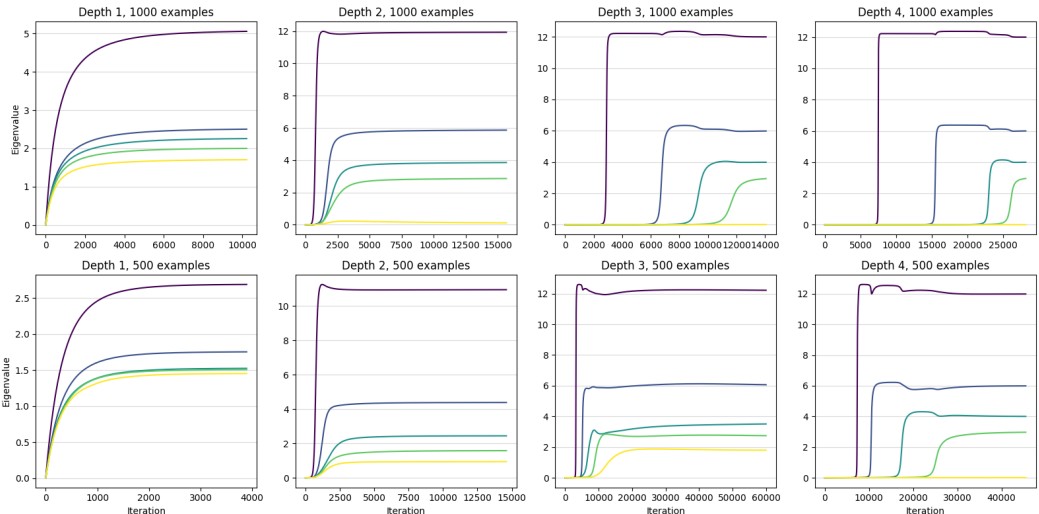

Figure 4: Evolution of the top-5 singular values of the deep matrix sensing model, with Gaussian initialization with variance such that the initial singular values are in expectation 1e-4. The model's size and data are in $\mathbb{R}^{50 \times 50}$. The columns correspond to different parameterization depths, while the rows correspond to different dataset sizes. In both cases the problem is over-determined, since the number of examples is smaller than the number of parameters. Since the original matrix is rank-4, we can recognize an unsuccessful recovery when all five singular values are nonzero, as seen clearly for both depth-1 plots.

sparse solutions - once the initialization conditions and depth are such that the learning phases are distinct, gradient descent finds the optimal rank-$i$ solution in every phase $i$. For these dynamics to successfully recover the optimal solution at every phase, the phases need to be far enough apart from each other to allow for the singular values and vectors to change before the next phase begins.

## F  INCREMENTAL LEARNING IN QUADRATIC NETWORKS

### F.1  PROOF OF THEOREM 5

**Theorem.** *Minimizing the quadratic network described and initialized as in* (7) *with gradient flow over the variance loss defined in* (2) *with Gaussian inputs leads to the following dynamical equations:*

$$\dot{\sigma}_i(t) = \sigma_i(t)(\sigma_i^* - \sigma_i(t))$$

*Where $\sigma_i$ and $\sigma_i^*$ are the $i$th singular values of $W$ and $W^*$, respectively, corresponding to the same singular vectors.*

*Proof.* Our proof will follow similar lines as the analysis of the deep matrix sensing model. Taking the expectation of the variance loss over Gaussian inputs for our model gives us:

$$
\begin{aligned}
\ell_{var}(W) =& \frac{1}{16}\mathbb{E}_x[(\langle W_*^T W_* - W^T W, xx^T \rangle)^2] - \frac{1}{16}\mathbb{E}_x[\langle W_*^T W_* - W^T W, xx^T \rangle]^2 \\
=& \frac{1}{8}\|W_*^T W_* - W^T W\|_F^2 + \frac{1}{16}Tr(W_*^T W_* - W^T W)^2 - \frac{1}{16}Tr(W_*^T W_* - W^T W)^2 \\
=& \frac{1}{8}\|W_*^T W_* - W^T W\|_F^2
\end{aligned}
$$

Following the gradient flow dynamics over $W$ leads to the following differential equation:

$$\dot{W} = \frac{1}{2}W(W_*^T W_* - W^T W)$$

We can now calculate the gradient flow dynamics of $W^T W$ using the chain rule:

$$W\dot{^T}W = W^T\dot{W} + \dot{W}^T W = \frac{1}{2}\Big(W^T W(W_*^T W_* - W^T W) + (W_*^T W_* - W^T W)W^T W\Big) \quad (10)$$

Now, under our initialization $W_0^T W_0 = \sigma_0 I$, we get that $W^T W$ and $W_*^T W_*$ are simultaneously diagonalizable at initialization by some matrix $U$, such that the following is true for diagonal $D$ and $D^*$:

$$D(0) = U^T W(0)^T W(0) U$$

$$D^* = U^T W_*^T W_* U$$

Multiplying equation (10) by $U$ and $U^T$ gives us the following dynamics for the singular values of $W^T W$:

$$\dot{D} = \frac{1}{2}\Big(D(D^* - D) + (D^* - D)D\Big) = D(D^* - D)$$

These matrices are diagonal at initialization, and remain diagonal throughout the dynamics (the off-diagonal elements are static according to these equations). We may now look at the dynamics of a single diagonal element, noticing it is equivalent to the depth-2 toy model:

$$\dot{\sigma}_i = \sigma_i(\sigma_i^* - \sigma_i)$$

$\square$

## F.2 DISCUSSION OF THE VARIANCE LOSS

It may seem that the variance loss is an unnatural loss function to analyze, since it isn't used in practice. While this is true, we will show how the dynamics of this loss function are an approximation of the square loss dynamics.

We begin by describing the dynamics of both losses, showing how incremental learning can't take place for quadratic networks as defined over the squared loss. Then, we show how adding a global bias to the quadratic network leads to similar dynamics for small initialization scales.

### F.2.1 DYNAMICAL DERIVATION

in the previous section, we derive the differential equations for the singular values of $W^T W$ under the variance loss:

$$\dot{\sigma}_i = \sigma_i(\sigma_i^* - \sigma_i)$$

We will now derive similar equations for the squared loss. The scaled squared loss in expectation over the Gaussian inputs is:

$$\begin{aligned}
\ell(W) =& \frac{1}{16}\mathbb{E}_x[(\langle W_*^T W_* - W^T W, xx^T\rangle)^2] \\
=& \frac{1}{8}\|W_*^T W_* - W^T W\|_F^2 + \frac{1}{16}Tr(W_*^T W_* - W^T W)^2
\end{aligned}$$

Defining $\Delta = W_*^T W_* - W^T W$ for brevity, the differential equations for $W$ become:

$$\dot{W} = \frac{1}{2}W\Delta + \frac{1}{4}Tr(\Delta)W$$

Calculating the dynamics of $W^T W$ after noting that it is simultaneously diagonalizable with $W_*^T W_*$ (as in the derivation for the variance loss) leads to the following differential equations for the singular values of $W^T W$:

$$\dot{\sigma}_i = \sigma_i\Big(\sigma_i^* - \sigma_i + \frac{1}{2}\sum_j (\sigma_j^* - \sigma_j)\Big)$$

We see that the equations are now coupled and so we cannot solve them analytically. Another issue is that for our initialization, all singular values have very similar dynamics at initialization due to the coupling. For example, values corresponding to small optimal singular values grow much faster than in the variance loss dynamics, due to the effect the large optimal singular values have on them.

We see from these equations that we shouldn't expect quadratic networks optimized over the squared loss to exhibit incremental learning behavior. We next show how adding a global bias to our model can help.

### F.2.2   Variance Loss Approximates the Squared Loss

To see how the variance loss can have dynamics resembling those of the squared loss, we will add a global (trainable) bias to our model. This means our model is now parameterized by $W \in \mathbb{R}^{d \times d}$ and a scalar $b \in \mathbb{R}$:

$$f_{W,b}(x) = \langle W^T W, xx^T \rangle + b$$

We may now analyze gradient flow of the squared loss. Following the same methods as before, this leads us to the following differential equations:

$$\dot{\sigma}_i = \sigma_i\Big(\sigma_i^* - \sigma_i + \frac{1}{2}\sum_j (\sigma_j^* - \sigma_j) + \frac{1}{2}(b^* - b)\Big)$$

$$\dot{b} = \sum_j (\sigma_j^* - \sigma_j) + b^* - b$$

Notice how, if $b$ is at it's optimum at a given time ($b = \sum_j (\sigma_j^* - \sigma_j) + b^*$), the dynamics of $\sigma_i$ align with those of the variance loss. Alternatively, when $b = b^*$, we recover the dynamics of the squared loss without the global bias term.

To convince ourselves that the dynamics of this model resemble those of the variance loss, we would need to explain why the global bias is at it's optimum "most of the time", such that the singular values don't change much during the times when it is not at it's optimum.

Observing the differential equations for $b$ and for $\sigma_i$, we see they are similar (if we ignore the $\sigma_i^* - \sigma_i$ value which doesn't change the order of magnitude of the entire expression when there haven't been many learning phases yet). The only difference being a multiplication by $\sigma_i$. This means that we may informally write:

$$\frac{\dot{\sigma}_i(t)}{\dot{b}(t)} \approx \sigma_i(t)$$

Since at initialization and until the learning phase of $\sigma_i$ takes place, we have $\sigma_i(t) \ll 1$, we see that the global bias optimizes much faster than the singular values for which the learning phase hasn't begun yet. This means these singular values will remain small during the times in which

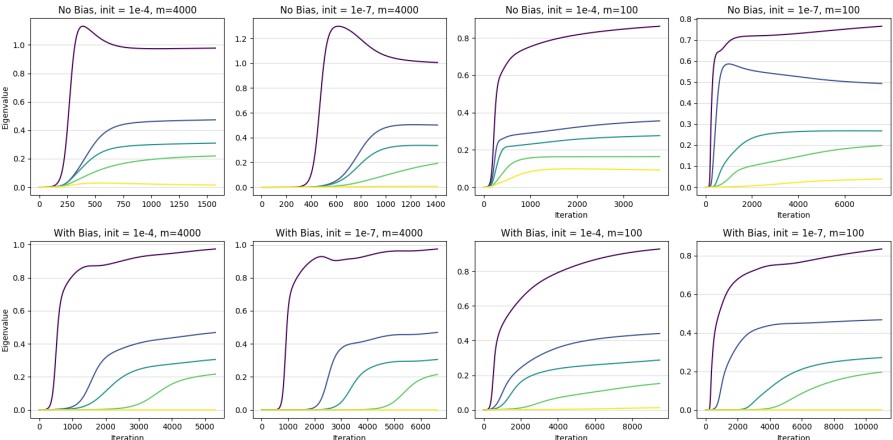

Figure 5: Quadratic model's evolution of top-5 singular values for a rank-4 labeling function. The rows correspond to whether or not a global bias is introduced to the model. The first two columns are for a large dataset (one optimal solution) and the last two columns are for a small dataset (over-determined problem). When a bias is introduced, it is initialized to it's optimal value at initialization. Note how without the bias, the singular values are learned together and there is over-shooting of the optimal singular value caused by the coupling of the dynamics of the singular values. For the small datasets, we see that the model with no bias reaches a solution with a larger rank. Once a global bias is introduced, the dynamics become more incremental as in the analysis of the variance loss. Note that in this case the solution obtained for the small dataset is the optimal low-rank solution.

the bias isn't optimal, and so incremental learning can still take place (assuming a small enough initialization).

Under these considerations, we say that the dynamics of the squared loss for a quadratic network with an added global bias resemble the idealized dynamics of the variance loss for a depth-2 linear model which we analyze formally in the paper. In figure 5 we experimentally show how adding a bias to a quadratic network does lead to incremental learning similar to the depth-2 toy model.

## G INCREMENTAL LEARNING IN CLASSIFICATION

### G.1 DIAGONAL NETWORKS

In section 4.3 we viewed our toy model as a special case of the deep diagonal networks described in Gunasekar et al. (2018), expected to be biased towards sparse solutions. Figure 6 shows the dynamics of the largest values of $\sigma$ for different depths of the model. We see that the same type of incremental learning we saw in earlier models exists here as well - the features are learned one by one in deeper models, resulting in a sparse solution. The leftmost panel shows how the initialization scale plays a role here as well, with the solution being more sparse when the initialization is small. We should note that these results do not defy the results of Gunasekar et al. (2018) (from which we would expect the initialization not to matter), since their results deal with the solution at $t \to \infty$.

### G.2 CONVOLUTIONAL NETWORKS - PRELIMINARIES

The linear circular-convolutional network of Gunasekar et al. (2018) deals with one-dimensional convolutions with the same number of outputs as inputs, such that the mapping from one hidden layer to the next is parameterized by $w_n$ and defined to be:

$$h_n[i] = \sum_{k=0}^{d-1} w_n[k]h_{n-1}[(i+k) \mod d] = (h_{n-1} \star w_n)[i]$$

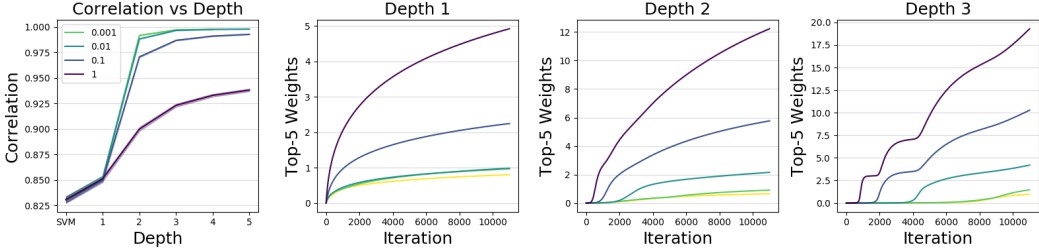

Figure 6: Incremental learning in binary classification. A model as in section 4.3 is trained over $200$ i.i.d. random Gaussian examples, where $d = 100$. The data is labeled by a weight vector with $4$ nonzero values, making the problem realizable with a sparse solution while the max-margin solution isn't sparse. The left panel describes the obtained solution's correlation with the sparse labeling vector for different depths and initializations. The results are averaged over 100 experiments, with shaded regions denoting empirical standard deviations. We see that depth-1 models reach results similar to the max-margin SVM solution as predicted by Gunasekar et al. (2018), while deeper models are highly correlated with the sparse solution, with this correlation increasing when the initialization scale is small. The other panels show the evolution of the absolute values of the top-5 weights of $\sigma$ for the smallest initialization scale. Note that as we increase the depth, incremental learning is clearly presented.

The final layer is a fully connected layer parameterized by $w_N \in \mathbb{R}^d$, such that the final model can be written in the following way:

$$f_\sigma(x) = ((((x \star w_1) \star w_2) \cdots) \star w_{N-1})^T w_N \tag{11}$$

Lemma 3 from Gunasekar et al. (2018) shows how we can relate the Fourier coefficients of the weight vectors to the Fourier coefficients of the linear model induced by the model:

**Lemma.** *For the circular-convolutional model as in* (11):

$$\hat{\sigma} = diag(\hat{w}_1) \cdots diag(\hat{w}_{N-1})\hat{w}_N,$$

*where for $n = 1, 2, ..., N, \hat{w}_n \in \mathbb{C}^d$ are the Fourier coefficients of the parameters $w_n \in \mathbb{R}^d$.*

This lemma connects the convolutional network to the diagonal network, and thus we should expect to see the same incremental learning of the values of the diagonal network exhibited by the Fourier coefficients of the convolutional network.

## G.3  CONVOLUTIONAL NETWORKS - EMPIRICAL EXAMINATION

In figure 7 we see the same plots as in figure 6 but for the Fourier coefficients of the convolutional model. We see that even when the model is far from the toy parameterization (there is no weight sharing and the initialization is with random Gaussian weights), incremental learning is still clearly seen in the dynamics of the model. We see how the inherent reason for the sparsity towards sparse solution found in Gunasekar et al. (2018) is the result of the dynamics of the model - small amplitudes are attenuated while large ones are amplified.

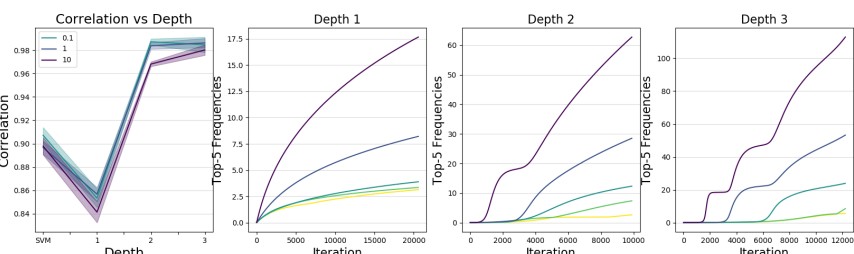

Figure 7: Incremental learning in convolutional networks. A model as in appendix G is trained over 200 i.i.d. random Gaussian examples, where $d = 100$. The weights are initialized randomly and the data is labeled by a weight vector with 4 nonzero frequencies, making the problem realizable with a sparse solution in the frequency domain. The left panel describes the obtained solution's correlation in the frequency domain with the sparse labeling vector for different depths and initializations. The results are averaged over 9 experiments, with shaded regions denoting empirical standard deviations. We see that depth-1 models reach results similar to the max-margin SVM solution, while deeper models are highly correlated with the optimal sparse solution. The other panels show the evolution of the amplitudes of the top-5 frequencies of $\sigma$ for the smallest initialization scale. Note that as we increase the depth, incremental learning is clearly presented.

