# OpenReview forum: "The Implicit Bias of Depth: How Incremental Learning Drives Generalization"
_ICLR.cc/2020/Conference — Accept (Poster)_

### Official Review · AnonReviewer2 · 2019-10-22
**Official Blind Review #2**

**Rating:** 6

**Review:**

*Contributions*
This paper deals with the theoretical study of the gradient dynamics in deep neural networks. More precisely, this paper define a notion of incremental learning for a particular learning dynamics and study how the depth of the network influence it. Then, the authors show two cases where it applies: matrix sensing, quadratic neural networks and provide intuitions on how it could also apply to linear convolutional networks.
This work leverage the framework and the proofs of Gidel et al. 2019 and Saxe et al. 2014 [1] to study the impact of depth in that sequential learning (that is the novelty of this work).

I really like the idea of studying the impact of the depth in the training dynamics. And the results (Thm2 and 3) are really interesting (but a bit hard to interpret in my opinion, see my questions)
Also, this work should make clearer that the results stated in Thm1 were already substantially presented in Saxe et al. 2013 ( eq. 17 and eq. 12 in [1])

Note that this paper is borderline regarding anonymity since it contains an acknowledgement section revealing the fundings of the authors (that could give enough information to identify the authors of this paper).

*Decision*
Weak accept: The key results in this work is Theorem 2 and it’s extension to discrete case Theorem 3. they seems really interesting: when $N > 2$ in order to observe sequential learning, the dependence in the eigengap for the initialization goes from polynomial ($N=2$) to polynomial.
Showing that result is interesting (even in this limited setting) because it theoretically shows (at least for these simple classes of problem) that deeper network perform a notion of incremental learning of components with non-prohibitively small initialization.
However, these results are very hard to read. They could be interpreted and simplified in that purpose. I think it would greatly improve the quality of this work.
I develop these points in the *questions* section of my review.

*Questions*
- The definition 1 is hard to interpret. For instance why do you need $t_i$ and $t_j$ since the functions $\sigma_i$ are increasing ? (note: the fact that these function are increasing is never mentioned in the paper but is key to talk about “incremental learning” and for your Definition 1 to make sense, since otherwise $\sigma_i$ could be “forgotten” without violating Definition 1) thus for any $t \in [t_i,t_j] $ we have $\sigma_i(t) \geq f \sigma_i^*$. Using only one time would make the definition easier to understand.
- The result presented in theorem 2, (and 3) are hard to interpret because of the many parameters that distract the reader to the main point. The dependence in s and f would be interesting if we would like to compute these bound in practice but I think that the interest of this work is in the distinction exponential versus polynomial (in $r$). Thus even though I think that a version with s and f is worth being in the appendix, a version with $s = f = ½$ would make the result statement and the discussion way clearer. (other question why restrict yourself to $s \in (0,½)$ and $f \in (¾,1)$?)
- In the theorem 3 who is $c$ ? who is $\sigma_1$ (the largest eigenvalue?) ?
- In theorem 3, I am very surprised that there is no notion of eigengap that restrict the size of $\eta$. for instance let us consider the three eigenvalues $\{2,2-\epsilon,1\}$ with epsilon very small. I think that this condition is implicitly appearing in A and B. Actually for a fixed $c$, if we do $\sigma_j^* \to \sigma_1^* = \sigma_i^*$ then we got $A = 1/B $ thus one of them is smaller than 1.
- You restrict yourself to a uniform initialization. Could you extend your results (and definitions) to non-uniform initialization (particularly initialization where
- Very small initialization is a big issue in practice because it induces very small gradient at the beginning of the training

- Figure 2, why is the time to learn those components increasing ? (i guess it is because of the $w^(2-2/N)$ that get smaller as $N$ increases. Isn’t it an issue in practice ? What is the sensitivity to the step size? in the discrete size (i.e. can we increase the step size to compensate the slower learning)?

*Minor remark*
- Saxe et al. 2013 as been accepted to ICLR in 2014 (would be better to cite the ICLR proceedings) see [1]

[1] Saxe et al. 2014 in ICLR url: https://openreview.net/forum?id=_wzZwKpTDF_9C


=== After rebuttal ===
I have read the authors' response.
I think the authors could have discussed more the interpretation of Theorem 3. in the revision.
But, I really like the main takeaway which is that there is a huge discrepancy between 2 and 3 layers in terms of dynamics.

I maintain my weak accept


**Experience Assessment:**

I have published one or two papers in this area.

**Review Assessment: Checking Correctness Of Derivations And Theory:**

I carefully checked the derivations and theory.

**Review Assessment: Checking Correctness Of Experiments:**

N/A

**Review Assessment: Thoroughness In Paper Reading:**

N/A

---

> ### Author Response · Authors · 2019-11-10
> **Response to review**
>
> We appreciate your thorough review of our paper. We will try to address your main concerns and questions:
>
> Regarding the clarity of the results, we agree they could have been be presented more clearly and thank you for the suggestions. We've incorporated some of them into the revision (see separate comment for a list of changes).
>
> You raised an interesting question about the dependency on the eigengap when fixing the learning rate. This dependency exists but is less obvious as it is in theorem 2 where the ratio of the values plays a clear role. In theorem 3 this ratio is represented within A and B:
> Assume as in your example that we fix c and take \sigma_{j}^{*} -> \sigma_{1}^{*}. In this case we indeed have that A=1/B and A is smaller than 1. This would lead the left hand side's exponent in theorem 3 to be negative while the right hand side's exponent is positive. In such a case, the LHS becomes larger than the RHS and there is no value of \sigma_{0} that can satisfy the condition, meaning it is impossible for incremental learning to take place under this learning rate and eigengap. This means that to have a satisfiable condition for \sigma_{0} when the eigengap is small, we may have to lower the learning rate to make A close to B.
> Theorem 2 shows that when the learning rate is infinitesimal and there is any positive eigengap, there is always a $\sigma_{0}$ for which incremental learning takes place. However, once the learning rate is finite, there are eigengaps in which no initialization allows for incremental learning.
>
> As for dealing with non-uniform initialization, our analysis method should be able to accommodate such a change and this sort of extension of our theory is possible. However, adding a non-uniform initialization would lead to even further parameters and more complicated results, which we see as more relevant for future work.
>
> Regarding the issue with very small initialization making the model hard to optimize - this is precisely the reason why depth is important for having a sparsity-inducing bias in practice. Only in deep models can we expect to have incremental learning in realistic initialization scales.
>
> As for the question about figure 2, this figure describes the dynamics of the solutions of the ODEs in theorem (1) and so the step size isn't relevant (and in general, it's effect on the optimization wasn't a big focus so far in our work).

---

### Official Review · AnonReviewer1 · 2019-10-23
**Official Blind Review #1**

**Rating:** 6

**Review:**

The paper studies the role of depth on incremental learning in several toy models for neural networks. In particular, they show that in these models, deep models require polynomially small initializations to exhibit incremental learning than shallow models. The paper is well written, and I think there are several interesting contributions.

The authors contribute analysis for non-asymptotically small initializations, and study an interesting role of depth in how small this initialization must be. Furthermore, they extend their results to several other models including matrix sensing, linear convnets, and classification.

I think nonetheless the paper suffers from a few issues. Some very important ones.

1) The authors study the role of depth on incremental learning, and exhibit how several models theoretically have this property. However, they do not study how incremental learning drives generalization. In the entire paper, the gradient flow is with respect to the *expected* loss, rather than the empirical loss. The research program of incremental learning for deep neural networks would show something like "Incremental learning exists when minimizing empirical loss", "Incremental learning and early stopping imply certain properties (like low capacity) on the resulting neural network" and "These properties imply low generalization error". However, the fact that the authors' models minimize the expected loss altogether a priori rules out the direct applicability of this result to explaining generalization. That is OK, in the sense that one could aim for these results to be modified and applied with empirical losses, and then a separate line of research could study how incremental learning bounds generalization error.
In this sense, I think the authors should take out the "how incremental learning drives generalization" since there is no study on generalization whatsoever, just how depth plays a role in incremental learning. An alternative title could be "How depth drives incremental learning." or something like that.

2) Another point is that all these models are very toy and mostly linear. That is OK again, but the introduction overclaims in this respect. The sentences "we characterize the effect of the model's depth [...] showing how deeper models allow for incremental learning in larger (realistic) initialization scales." and "Once incremental learning has been defined and characterized for the toy model, we generalize our results theoretically and empirically for larger models". This makes it seem that results apply to realistic settings, which is really far from true. I'm not expecting realistic results, this is a nascent theory, but I am expected the claims made to be validated and not misleading.

3) In section 2.2, sigma(t) for N-> \infty is undefined, and the proof for this result is missing (only for finite N appears). In particular, it is not clear if sigma(t) for N -> \infty is obtained by a) taking limit N -> \infty in the ODE of equation (8), and then finding the solution of this limiting ODE, or b) finding \sigma(t) for equation (8) on finite N and then taking limit N -> \infty of the solution. I.e. there are two potentially different ways to define \sigma(t) for N -> \infty which are solving the ODE and then taking limit or taking limit and then solving the limiting ODE. The definition of \sigma(t) for N -> \infty is completely missing so I have no way to assess the validity of this result.

A small pet peeve: when writing math, try to avoid using symbols like \forall and \exists unless you're writing a logic paper. Instead, try to write equation 2 like $$\sigma_i = w_i^N \quad \text{for all i = 1, \dots, d} $$, which reads a lot nicer. Also, avoid assigning equation numbers to equations you never reference.

**Experience Assessment:**

I have read many papers in this area.

**Review Assessment: Checking Correctness Of Derivations And Theory:**

I assessed the sensibility of the derivations and theory.

**Review Assessment: Checking Correctness Of Experiments:**

I assessed the sensibility of the experiments.

**Review Assessment: Thoroughness In Paper Reading:**

I read the paper at least twice and used my best judgement in assessing the paper.

---

> ### Author Response · Authors · 2019-11-10
> **Response to review**
>
> Thank you for the detailed review of our paper.
>
> As we understand it, the main issues are with us overclaiming our results, and we accept some of this criticism.
> Regarding our introduction where it can be interpreted that the models we analyze are "realistic", we see the models we analyze as non-trivial but concede that the wording could be interpreted as a claim for deep nonlinear networks, which our analysis by no means covers. Our revision fixes this, clarifying which models we generalize to.
> Regarding the claim that our paper deals with generalization while the analysis is for the expected loss (meaning the theoretical study is about the path to the optimum and not the specific optimum chosen) - it is true that our theorems do not deal with empirical losses, but our experiments seen in figure 1 and the appendices strongly suggest that the dynamics we analyze for the expected loss are exhibited for different empirical losses and for small datasets. The bias towards sparsity, which leads to generalization when the labeling function is itself sparse, appears to be caused by the combination of gradient descent and a deep parameterization which we analyze, and is consistent for many deep models.
>
> As for the third issue, the limit we take is for the ODE in equation (8), which leads to the equation $\dot{\sigma}=\sigma^{2}(\sigma^{*}-\sigma)$, which is solvable like the ODEs for finite N. This result also appears in Saxe et al (2013). Our revision treats this case more clearly in the proof of theorem 1.

---

### Official Review · AnonReviewer3 · 2019-11-10
**Official Blind Review #3**

**Rating:** 6

**Review:**

This paper studies the phenomenon of incremental learning in several deep models. It starts with analyzing the optimization dynamics of a toy model, and showing that it follows incremental learning, a notion defined clearly in the paper. In particular, it shows that depth affects the strength of incremental learning in the sense that when the depth of the model is increased (especially when going from N=2 to N=3), the maximal initialization value with which incremental learning can occur is increased. In this sense, deeper models experience incremental learning more easily. The paper then moves on to other “deep linear“ models, including matrix sensing, one-hidden-layer quadratic neural networks and diagonal/convolutional linear neural networks, derives ODEs for the evolution of the singular values in the learned models, which is argued to also lead to incremental learning.

I would recommend a “weak accept” for this paper. The nice contributions include a clear definition of incremental learning, results showing the depth’s effect on incremental learning as well as extensions to several other models. My main question is regarding the relevance of the toy model to more realistic models, as I will discuss below, and I’d love to hear more about the authors’ thoughts on this.

Besides being linear, another important simplification of the toy model is that there is no interaction among the hidden units, which is rather crucial for ordinary neural networks. I am curious to what extent the authors think this simplification matters for incremental learning. It’s nice that similar analysis can be extended to other settings including matrix sensing, quadratic NNs and linear diagonal/convolutional NNs. But it seems that there is no theorem analogous to theorems 2 and 3 for those models, and I am curious why.

The qualitative transition from N=2 to N=3 in the toy model is interesting. Is there a more intuitive explanation for it? Also, from Figure 2, it seems hard to say whether there really is a qualitative change between N=2 and N=3.

Some other suggestions for improvement:
1. It may be helpful to somehow visualize the bounds obtained in theorems 2 and 3.
2. Some typos:
1) “it’s” should be “its” in the last paragraph of section 2.
2) ”effect” should be “affect” in the first paragraph of section 3.

**Experience Assessment:**

I do not know much about this area.

**Review Assessment: Checking Correctness Of Derivations And Theory:**

I assessed the sensibility of the derivations and theory.

**Review Assessment: Checking Correctness Of Experiments:**

I assessed the sensibility of the experiments.

**Review Assessment: Thoroughness In Paper Reading:**

I read the paper at least twice and used my best judgement in assessing the paper.

---

> ### Author Response · Authors · 2019-11-11
> **Response to Review**
>
> Thank you for your thoughtful review. We will try to answer any open questions:
>
> Regarding your comments about the interaction of the layers in realistic models and lack thereof in our toy model, you touch on an interesting open question.
> First, we would like to clarify that even when interaction does occur between hidden neurons, there can be theoretical results like our theorem 2 and 3 - the deep matrix sensing model we analyze parameterizes the matrix as a product of N matrices, and these N matrices interact similarly to "regular" deep linear networks. While perhaps not stated clearly enough, our theorem 4 shows that under the right conditions, the singular values of the matrix sensing model have the same dynamics as our toy model, and this means that under these conditions theorems 2 and 3 apply for matrix sensing. The same goes for quadratic networks (although the conditions are admittedly not as natural). Also, when we look at the original work of Saxe et al. [1], we see derivations of very similar dynamical equations for the singular values of deep linear networks under the squared loss - slight adaptations to our analysis would lead to an incremental learning result for this model as well.
> However, things are not as simple as one would hope - if we look at classification tasks (with exponential-tailed losses), we see that the interaction between layers has a different effect. In [2], we see that deep linear networks trained on separable data converge to the max-margin solution regardless of the depth of the network, while when there is no interaction between the layers (diagonal/convolutional models), deeper models tend towards sparser solutions (which we claim is caused by a form incremental learning dynamics, but only show empirical evidence of this).
> It seems that there is more to incremental learning than the depth and initialization of the model - the loss function and exact parameterization play a role. The interactions between the parameterization and depth of the model, the initialization and the loss function are non-trivial and an interesting research direction.
>
> As for the strong difference between depth-2 and depth-3 models, we're afraid we can't really offer an intuitive explanation. When looking at figure 2, the qualitative difference that we were referring to is when comparing different initializations. The top row ($\sigma_{0}=0.1$) has both depths not really exhibiting incremental learning, but once we decreases the initialization in the second row ($\sigma_{0}=0.001$), we see the difference - the depth-2 model hardly changes its dynamics (still no strong incremental learning) while the depth-3 model exhibits very strong incremental learning dynamics.
>
> [1] - Andrew M Saxe, James L McClelland, and Surya Ganguli. Exact solutions to the nonlinear dynamics of learning in deep linear neural networks. ICLR, 2014.
> [2] - Suriya Gunasekar, Jason D Lee, Daniel Soudry, and Nati Srebro. Implicit bias of gradient descent on linear convolutional networks. In Advances in Neural Information Processing Systems, pp. 9461–9471, 2018

---

### Author Response · Authors · 2019-11-10
**Summary of changes in the revision**

Thank you for your reviews!
We've uploaded a revision of the paper following the remarks of both reviewers. The following is a summary of the changes:
1. We've clarified at the end of the introduction which models we generalize our toy model to, as to not overclaim our results.
2. More explicit credit has been given to the work of Saxe et al. when presenting theorem 1.
3. The case of N->\infty is now treated in the proof of theorem 1.
4. The definition of incremental learning was simplified (Definition 1).
5. The parameters c and \sigma_{1} are more clearly explained in theorem 3.

We look forward to further comments and discussions.

---

### Decision · Program_Chairs · 2019-12-19

**Decision:**

Accept (Poster)

**Comment:**

The paper studies the role of depth on incremental learning, defined as a favorable learning regime in which one searches through the hypothesis space in increasing order of complexity. Specifically, it establishes a dynamical depth separation result, whereby shallow models require exponetially smaller initializations than deep ones in order to operate in the incremental learning regime.

Despite some concerns shared amongst reviewers about the significance of these results to explain realistic deep models (that exhibit nonlinear behavior as well as interactions between neurons) and some remarks about the precision of some claims, the overall consensus -- also shared by the AC -- is that this paper puts forward an interesting phenomenon that will likely spark future research in this important direction. The AC thus recommends acceptance.